# A Preliminary Assessment of the Nutraceutical Potential of Acai Berry (*Euterpe* sp.) as a Potential Natural Treatment for Alzheimer’s Disease

**DOI:** 10.3390/molecules27154891

**Published:** 2022-07-30

**Authors:** Maryam N. ALNasser, Ian R. Mellor, Wayne G. Carter

**Affiliations:** 1Department of Biological Sciences, College of Science, King Faisal University, P.O. Box No. 400, Al-Ahsa 31982, Saudi Arabia; mnan1@hotmail.com; 2School of Life Sciences, Faculty of Medicine and Health Sciences, University of Nottingham, Nottingham NG7 2RD, UK; ian.mellor@nottingham.ac.uk; 3School of Medicine, Royal Derby Hospital Centre, University of Nottingham, Derby DE22 3DT, UK

**Keywords:** acai berry, Alzheimer’s disease, antioxidant, cholinesterase inhibitors, *Euterpe oleraceae*, nutraceuticals

## Abstract

Alzheimer’s disease (AD) is characterised by progressive neuronal atrophy and the loss of neuronal function as a consequence of multiple pathomechanisms. Current AD treatments primarily operate at a symptomatic level to treat a cholinergic deficiency and can cause side effects. Hence, there is an unmet need for healthier lifestyles to reduce the likelihood of AD as well as improved treatments with fewer adverse reactions. Diets rich in phytochemicals may reduce neurodegenerative risk and limit disease progression. The native South American palm acai berry (*Euterpe oleraceae*) is a potential source of dietary phytochemicals beneficial to health. This study aimed to screen the nutraceutical potential of the acai berry, in the form of aqueous and ethanolic extracts, for the ability to inhibit acetyl- and butyryl-cholinesterase (ChE) enzymes and scavenge free radicals via 2,2-diphenyl-1-picryl-hydrazyl-hydrate (DPPH) or 2,2′-azino-bis-3-ethylbenzthiazoline-6-sulphonic acid (ABTS) assays. In addition, this study aimed to quantify the acai berry’s antioxidant potential via hydrogen peroxide or hydroxyl scavenging, nitric oxide scavenging, lipid peroxidation inhibition, and the ability to reduce ferric ions. Total polyphenol and flavonoid contents were also determined. Acai aqueous extract displayed a concentration-dependent inhibition of acetyl- and butyryl-cholinesterase enzymes. Both acai extracts displayed useful concentration-dependent free radical scavenging and antioxidant abilities, with the acai ethanolic extract being the most potent antioxidant and displaying the highest phenolic and flavonoid contents. In summary, extracts of the acai berry contain nutraceutical components with anti-cholinesterase and antioxidant capabilities and may therefore provide a beneficial dietary component that limits the pathological deficits evidenced in AD.

## 1. Introduction

Approximately one million people are affected by neurodegenerative diseases (NDDs) in the United Kingdom, and more than 50 million people suffer worldwide from dementia [1,2]. These diseases are of global concern as their prevalence is continuously increasing and is associated with an enormous socioeconomic burden [3]. One widespread NDD is Alzheimer’s disease (AD), which accounts for 60–80% of dementia cases and for which the annual care costs for the USA alone have been estimated at around $244 billion [4,5]. This progressive degenerative brain disease can be identified by clinical and histopathological hallmarks [6]. The early-stage clinical symptoms of AD involve memory loss for new events or conversations, lack of concern, and depression, with advanced symptoms including behavioural changes, the inability to communicate or speak, difficulty swallowing and walking, and a loss of a sense of direction [7]. Histopathological damage to the brain includes the deposition of plaques composed of β-amyloid (Aβ) in the extracellular space around neurons and intraneuronal threads of hyperphosphorylated tau protein (p-tau) that are termed neurofibrillary tangles (NFTs) [6,7,8].

The pathogenicity of AD is complex, with pathomechanisms that include oxidative stress, excitotoxicity, impaired mitochondrial function, and disrupted cholinergic signalling, as well as toxic peptide and protein accumulations [8,9,10,11,12,13,14,15,16,17]. Multiple risk factors contribute to the development of AD, with the greatest risk factor being age itself [18]. Other risk factors include familial genetics [19,20] and sporadic mutations, as well as dietary influences and other environmental factors, such as exposures to metals, pesticides, solvents, and other neurotoxic agents [8,21,22,23].

The currently approved drugs for the symptomatic relief of mild-to-moderate AD are acetylcholinesterase (AChE) inhibitors such as galantamine, rivastigmine, and donepezil, as well as the N-methyl-D-aspartate receptor antagonist memantine [7,24,25]. Even though these medications can potentially slow disease progression, none can prevent or stop the course of the disease [26,27,28]. In addition, these treatments can cause adverse side effects, including headaches, nausea, vomiting, diarrhoea, dizziness, confusion, and cardiac arrhythmias [25,29,30]. The actions of drugs such as cholinesterase inhibitors (ChEIs) and memantine involve the modulation of the levels and activity of neurotransmitters, such as acetylcholine (ACh) and glutamate (Glu), respectively [24,25]. However, these drugs were not originally developed to resolve other pathological mechanisms implicated in the development and/or progression of AD, such as tissue damage from oxidative stress, which has been detected in *post-mortem* brain tissue from AD patients [16,31,32,33,34,35,36]. Hence, a search continues for other novel treatment strategies with additional activities that target cholinergic deficits as well as other elements of AD pathology, with the expectation of reduced side effects.

Nutraceuticals encompass dietary substances that have physiological benefits or confer resistance to or protection from the development of diseases. The utilisation of nutraceuticals and dietary supplements is expanding, with the proposed benefit of combatting a number of diseases, including neurodegeneration [37,38]. Specifically, the dietary intake of bioactive food compounds has been linked with the prevention of age-dependent memory and cognition decline [39,40,41]. Clinical trials on patients with mild cognitive impairment and healthy people of similar age have indicated that the regular intake of fruits, such as grapes and berries, can have positive effects on cognition [42,43,44,45]. Certainly, there is an actual or perceived assumption that a diet rich in fruit and vegetables is one that is considered ‘healthy’ and has the potential for the improvement or prevention of cognitive decline [46,47,48].

The *Euterpe* genus has roughly 28 species that are found across the Amazon basin in Central and South America [49]. Two species, *E. oleracea* and *E. precatoria*, are widely marketed for their edible fruit [49]. Within the Amazonas River Basin, *E. precatoria* is a native variety and is commonly known as “acai-do-amazonas” [49], whereas *E. oleracea*, or “acai-do-pará,” is widely spread throughout the Amazon River estuaries, as well as in Guyana, Venezuela, and the Brazilian estates of Pará, Maranho, Tocantins, and Amapá [49,50]. These species are palm trees with small, rounded, and clustered fruits approximately 0.9–1.3 cm in diameter [49]. The mature fruit is dark purple and globose and has one seed covered with a 2 mm juicy mesocarp layer [49]. Traditional medicines have utilised different parts of this plant to treat a number of illnesses, including fever, gastrointestinal and skin conditions, pain, and infectious diseases [49,51].

Collectively, there have been several clinical, animal, and cell-based studies that have reported the potential health benefits of acai berries, including their antioxidant activity [52,53,54]. Therefore, this study aimed to analyse the nutraceutical potential of *Euterpe* sp. (in the form of aqueous and ethanolic extracts) for possible development as an AD treatment via the ability to inhibit AChE and butyrylcholinesterase (BuChE); the study also aimed to further delineate its antioxidant capabilities.

## 2. Results

### 2.1. Acai Aqueous Extract Is a Cholinesterase Inhibitor

The acai berry aqueous extract significantly inhibited AChE activity in a concentration-dependent manner over concentrations ranging from 1 × 10^−5^ µg/mL to 0.01 µg/mL, but no further inhibition of AChE activity was observed at concentrations higher than 0.01 µg/mL (Figure 1A). The concentration of acai aqueous extract that produced a 50% inhibition (IC_50_) of AChE activity was estimated as 13.8 µg/mL using non-linear regression. In contrast, the acai berry ethanolic extract showed a limited inhibition (1–19%) of AChE, and only at the relatively high extract concentrations of 100 and 1000 µg/mL were there inhibition levels of 16% and 19%, respectively; however, this was not statistically significant (Figure 1B).

The acai berry aqueous extract was also primarily a concentration-dependent inhibitor of BChE activity over concentrations ranging from 1 × 10^−5^ µg/mL to 0.01 µg/mL (Figure 1C). However, similar to the inhibition of AChE, a point of saturation was detected, with the further inhibition of BuChE observed only at the highest concentration examined, i.e., 1000 µg/mL (43.6%) (Figure 1C). The concentration of acai aqueous extract that inhibited BuChE activity by 50% (IC_50_) was estimated as 6378 µg/mL using non-linear regression. Similar to the inhibition of AChE, the acai berry ethanolic extract also showed a limited inhibition of BuChE, with only marginal inhibition levels (12–22%) at concentrations of 0.01 µg/mL or higher, and these effects were not statistically significant (Figure 1D).

### 2.2. Acai Aqueous and Ethanolic Extracts Exhibit 2,2-Diphenyl-1-picrylhydrazyl (DPPH) Free Radical Scavenging Activity

The ability of aqueous and ethanolic extracts of acai berry to act as free radical scavengers was assessed using a DPPH assay. Both extracts displayed DPPH radical scavenging abilities and had similar concentration-dependent curves, although the aqueous extract had a lower percentage of activity than the ethanolic extract over the concentration range of 1000–4000 µg/mL (Figure 2A). The antioxidants α-tocopherol (vitamin E), L-ascorbic acid (vitamin A), and gallic acid all displayed higher free radical scavenging over the concentration range of 1000–4000 µg/mL. The concentration of each of the agents that produced a 50% inhibition of free radical levels (IC_50_) was calculated by non-linear regression as 11.550 mg/mL for the acai aqueous extract and 791 µg/mL for the ethanolic extract. By comparison, for α-tocopherol, L-ascorbic acid, and gallic acid, the IC_50_ values were 50 µg/mL, 115 µg/mL, and 8 µg/mL, respectively. At the lower end of the concentrations examined (from 0.01 µg/mL to 10 µg/mL), the aqueous extract displayed a similar antioxidant capability to α-tocopherol, and the ethanolic extract surpassed that of α-tocopherol and gallic acid (Figure 2B).

### 2.3. Acai Aqueous and Ethanolic Extracts Exhibit 2,2′-Azino-bis-3-ethylbenzthiazoline-6-sulphonic acid Radical Cation (ABTS^•+^) Scavenging Activity

Acai berry aqueous and ethanolic extracts exhibited ABTS^•+^ scavenging activity in a concentration-dependent manner that was approximately linear over the concentration range of 1–1000 µg/mL (Figure 3). The acai ethanolic extract displayed a greater antioxidant capacity than either L-ascorbic acid or α-tocopherol, with an IC_50_ of 461.6 µg/mL compared to 690 µg/mL and an estimated 1270 µg/mL for L-ascorbic acid and α-tocopherol, respectively. Gallic acid exhibited the greatest scavenging activity, with an IC_50_ of 8 µg/mL. The acai aqueous extract showed the lowest antioxidant capacity, with an estimated IC_50_ of 30.541 mg/mL.

### 2.4. Acai Aqueous and Ethanolic Extracts Exhibit Hydrogen Peroxide (H_2_O_2_) Scavenging Activity

Both acai extracts displayed moderate but concentration-dependent H_2_O_2_ scavenging activity (Figure 4), with an inhibition percentage of 20–30% and estimated IC_50_ values of 7.803 mg/mL for the aqueous extract and 1.479 mg/mL for the ethanolic extract. α-Tocopherol and gallic acid were more potent H_2_O_2_ radical scavengers, with IC_50_ values of 676 µg/mL and 737 µg/mL, respectively.

### 2.5. Acai Aqueous and Ethanolic Extracts Exhibit Hydroxyl Radical (^•^OH) Scavenging Activity

Both acai extracts exhibited hydroxyl radical (^•^OH) scavenging activity in a concentration-dependent manner, as shown in Figure 5. In comparison to the acai aqueous extract, the acai ethanolic extract showed higher antioxidant action, with an IC_50_ of 946 µg/mL, while the IC_50_ of acai aqueous extract was estimated as 11.604 mg/mL. Gallic acid was a potent ^•^OH radical scavenger, with an IC_50_ of 0.7 µg/mL.

### 2.6. Acai Aqueous and Ethanolic Extracts Exhibit Nitric Oxide (^•^NO) Scavenging Activity

The percentage of ^•^NO scavenging increased in proportion to the concentration of the acai extracts, as displayed in Figure 6. The acai ethanolic extract exhibited greater inhibition activity than the aqueous extract; it had an estimated IC_50_ of 4.544 mg/mL, whereas the estimated IC_50_ of the aqueous extract was 12.932 mg/mL. The standard BHA displayed a higher scavenging ability at lower concentrations, but these reached saturation such that the estimated IC_50_ was 135.437 mg/mL, which was higher than either of the two acai extracts.

### 2.7. Acai Aqueous and Ethanolic Extracts Exhibit Lipid Peroxidation Inhibitory Activity

The concentration-dependent inhibition of lipid peroxidation was observed after incubation with either the acai berry aqueous or ethanolic extract, as shown in Figure 7. Both extracts displayed relatively moderate anti-lipid peroxidation in comparison with BHA. The acai aqueous extract displayed greater antioxidant activity than the ethanolic extract, with estimated IC_50_ values of 4.862 mg/mL and an estimated IC_50_ of 438.8 mg/mL, respectively; BHA had an IC_50_ of 4 µg/mL. At the highest concentration examined, i.e., 1000 µg/mL, the inhibition of lipid peroxidation was 36.5% ± 0.51 and 26.8% ± 1.60 for the acai aqueous and ethanolic extracts, respectively, and 82.8% ± 0.16 for BHA.

The IC_50_ values for each of the cholinesterase and antioxidant assays are included in Table 1.

### 2.8. Acai Aqueous and Ethanolic Extracts Exhibit Reducing Power Activity

The direct reduction of Fe[(CN)_6_]_3_ to Fe[(CN)_6_]_2_ provides a determination of the reducing capacity of a plant compound [55]. The reducing capacity of acai berry aqueous and ethanolic extracts was concentration-dependent but relatively low compared to L-ascorbic acid over the 1000–8000 µg/mL concentration range, as shown in Figure 8. The acai ethanolic extract exhibited more antioxidant capacity than the aqueous extract. However, at the lower concentrations of 0.001–10 µg/mL the reducing capacities of the two acai extracts were similar and matched that of L-ascorbic acid (results not shown).

### 2.9. Total Phenolic and Total Flavonoid Content of Acai Berry Extracts

The total phenolic content (TPC) and total flavonoid content (TFC) of the acai aqueous and ethanolic extracts were quantified and are included in Table 2. Acai ethanolic extract displayed higher levels of phenolic and flavonoid contents.

## 3. Discussion

The present study evaluated the nutraceutical and, hence, the therapeutic potential of acai berry extracts via their ability to act as ChEIs; also assessed were the additional benefits of free radical scavenging and antioxidant activities. The current first-line treatment for mild-to-moderate AD is to treat the cholinergic deficit experienced by AD patients via a transient inhibition of AChE to increase the signal longevity of the neurotransmitter acetylcholine (ACh) [24,25,56]. However, ChEI treatment can induce adverse reactions and only addresses one component of the AD disease pathology (insufficient ACh levels), whereas multiple elements of cellular dysfunction may contribute to the disease, including oxidative stress [25,30,31,32,33,34,35,36,57]. Hence, there is an unmet need to tackle disease aetiology, for example, through a multipronged treatment strategy [57,58]. Indeed, animal models of AD have demonstrated improvements in cognitive function and behavioural defects after antioxidant therapy [59,60].

The aqueous extract from the acai pulp contained an agent(s) capable of inhibiting both AChE (estimated IC_50_ of 13.8 µg/mL) and, to a lesser extent, BuChE (estimated IC_50_ of 6.378 mg/mL). Interestingly, the agent(s) binding the cholinesterases presumably reached a point of saturation at an approximate concentration of 0.01 µg/mL, such that further enzymatic inhibition was limited. This may result from finite binding at the esteratic and/or peripheral binding sites of AChE or BuChE [61,62,63,64]; this question can be probed further once the active agent(s) is/are purified. By comparison, the ethanolic extract displayed minimal anti-AChE and anti-BuChE activities; there were only minor reductions in activity evident at the highest concentrations of extract, and these did not reach significance. This may reflect the solvation of the agent(s) within water alone (rather than ethanol) since water is a more polar solvent than ethanol. Ultimately, a range of extraction methods and solvents may be needed to isolate active agents such as polyphenols, with the solvation of specific phytochemical(s) governed by the polarity of the solute of interest [65].

Specifically, for acai berry phytochemical extractions, an independent study reported that water as a solvent produced the highest yields of polyphenols and flavonoids as compared with methanol and ethanol alone [66], although this may be improved further if a hydroalcoholic extraction is undertaken (50% ethanol) [67]. Herein, the benefit of an aqueous extraction as a method for the possible isolation of soluble ChEI(s) and their future purification and identification was evident.

Chemicals able to target and simultaneously inhibit both AChE and BuChE, rather than AChE alone, may offer improved clinical efficacy to combat the increased levels of BuChE in AD patients, with a reduction in side effects [68,69,70,71]. However, neither of the drugs currently approved by the US Food and Drug Administration (FDA), namely donepezil and galantamine, are potent BuChE inhibitors (BuChE IC_50_ values of 5 µM and 12.6 µM, respectively), whereas rivastigmine, which was originally extracted from a medicinal plant, is a relatively potent AChE and BuChE inhibitor (IC_50_ values of 4 and 13 nM, respectively) [72,73]. It will be of interest in future studies to determine whether the same agent(s) within the acai aqueous extract is/are responsible for inhibiting both AChE and BuChE, and to determine the relative potency of the inhibitor(s). A comparison of the anti-AChE activity of the extracts/fractions of 54 plant species used guidelines that considered an IC_50_ < 20 µg/mL as high potency, moderate potency as an IC_50_ > 20 µg/mL but < 200 µg/mL, and low potency as an IC_50_ > 200 µg/mL but < 1000 µg/mL [72]. The cut-offs for potency related to the average IC_50_ value for galantamine, reported in the literature as approximately 2 µM (or 0.575 µg/mL), multiplied by a factor of 10 [74]. Accordingly, the aqueous extract of acai fruits, with an IC_50_ of 13.8 µg/mL, was an extract of high potency and therefore has potential use as an effective ChEI.

In addition to ChEI activities, the acai extracts displayed useful radical scavenging and antioxidant activities; they were able to scavenge DPPH, ABTS, ^•^OH, H_2_O_2_, and ^•^NO free radicals, and they exhibited ferric ion reduction. Similarly, other independent studies have reported the high antioxidant capacity of the acai berry against superoxide (O^2•−^) and peroxyl radicals (RO_2_) [75]. Acai also displays useful neuroprotective activity, and it can prevent rotenone-induced oxidative damage [76]. Acai pulp also reduced nitrite radicals (NO^2−^) in mouse brain BV-2 microglial cells in vitro [53]. Acai flower and spike fractions (as well as the fruit) also contain agents able to inhibit nitric oxide production [77]. In addition, in a pilot study with human volunteers, the antioxidant capacity of plasma was elevated by 2.3- and 3-fold after the consumption of acai juice and pulp, respectively [78]. It is also promising to note that an in vivo study demonstrated that an acai-rich diet reduced markers of oxidative stress in brain regions of aged rats [79].

The current study also demonstrated that acai extracts have anti-lipid peroxidation effects, in support of studies in vitro [75] and in vivo [54]. The significant inhibition of lipid peroxidation was also detected post-consumption of a juice blend (of which acai was the predominant ingredient) in a pilot study with human volunteers [52].

Research has considered the chemical composition of the acai berry and its antioxidant potential and has detected the presence of numerous polyphenols and flavonoids, such as anthocyanins [52,76,77,78,80,81] (refer also to Appendix A). Phytochemicals such as these may provide the basis for the acai extract to neutralise free radicals and potentially limit oxidative stress, such as that associated with AD aetiology, but further in vitro and in vivo studies are required to confirm the potential use of acai berry extracts as a treatment option for AD. Ultimately, a diet that incorporates acai berries may provide the ongoing benefit of a diet rich in antioxidants along with the possibility of sustained cholinergic signalling that may limit the likelihood of developing or indeed the progression of NDDs such as AD.

## 4. Materials and Methods

### 4.1. Chemicals and Reagents

All chemicals were purchased from Sigma (Poole, UK) unless otherwise specified.

### 4.2. Preparation of Ethanolic and Aqueous Extracts of Acai Berry (Euterpe oleracea)

An ethanolic extract of acai berry was prepared by the maceration of 300 mg/mL of commercially available freeze-dried acai pulp and skin powder purchased from NaturaleBio (Organic product under EU Directive 834/2007, purchased via Amazon.co.uk) in 70% ethanol for 48 h. The macerated sample was shaken 3 times daily to assist solvation and then filtered using a bottle-top filter. Filtrates were dried at 45–50 °C for 24 h in a water bath to obtain the ethanolic dry extracts [76,82]. The aqueous extract (10 mg/mL) was prepared using methods as described by Wong et al. (2013) [83]. The freeze-dried acai pulp and skin powder was weighed and extracted by dissolving in phosphate-buffered saline (PBS) and by vigorous vortexing. The extract was centrifuged at 400 rpm and filtered using a 0.20 µm syringe to obtain a clear solution.

### 4.3. Cholinesterase Activity Assessments

Based on the method of Ellman et al. (1961) [84], the ability of the acai berry extracts to inhibit the activity of AChE and BuChE was assessed in a 96-well microtiter plate. Ten µL of acai aqueous or ethanolic extracts (concentration range from 1 × 10^−6^ µg/mL to 1000 µg/mL) was mixed with 150 µL of 0.38 mM 5,5-dithio-bis-(2-nitrobenzoic acid) (DTNB), 3 µL of 0.5 U/mL AChE enzyme from Electrophorus electricus (electric eel) (Sigma, C3389, Irvine, UK) or BuChE enzyme from equine serum (Sigma, C75120, Irvine, UK), and 43 µL of phosphate buffer (Gibco™ PBS, pH 7.4, ThermoFisher, Stafford, UK). Samples were incubated for 20 min at room temperature; then, the reaction was initiated by the addition of 4 µL of 35 mM of acetylthiocholine iodide (ATCI) substrate for AChE or butyrylthiocholine iodide (BTCI) substrate for BuChE, and the absorbance was measured at 412 nm every 30 s for 5 min using a Varioskan™ LUX multimode microplate reader (ThermoFisher, UK). Reagent blanks were performed in the absence of AChE or BuChE. The positive (inhibitor) control for AChE assays was an organophosphate pesticide, azamethiphos (QMX Laboratories Ltd., Thaxted, UK) at 5 mM, capable of the irreversible inhibition of AChE [85]. For BuChE assays, ethopropazine hydrochloride (QMX Laboratories Ltd., Thaxted, UK) at 5 mM was used as a recognised inhibitor of BuChE [86]. The percentage of AChE or BuChE activity remaining after incubation with acai extracts was calculated relative to the enzyme only (the negative control), which was designated as 100% enzymatic activity. The acai extract concentrations producing 50% inhibition (IC_50_) of AChE or BuChE activity were determined. The assays were performed in duplicate for at least three independent experiments, after which a mean was calculated.

### 4.4. 2,2-Diphenyl-1-picrylhydrazyl (DPPH) Free Radical Scavenging Activity

The antioxidant capacities of acai aqueous and ethanolic extracts over the concentration range of 0.01–4000 µg/mL were evaluated by monitoring the ability to reduce the stable free radical di(phenyl)-(2,4,6 trinitrophenyl)iminoazanium (DPPH) according to a previous publication [87]. DPPH was dissolved in ethanol at a final concentration of 0.1 mM, then 160 µL of this solution was added to 20 µL of either acai extract or L-ascorbic acid, α-tocopherol, or gallic acid as positive control antioxidants, and the material was mixed with 20 µL of distilled water. Antioxidant standards were evaluated using the same concentration range as the acai extracts. The mixture was incubated for 40 min in the dark at 37 °C, and then the absorbance was read using a Varioskan™ LUX multimode microplate reader (ThermoFisher, Stafford, UK) at 517 nm as an endpoint measurement. Antioxidant activity was calculated as a percentage of the DPPH radical scavenging activity according to the following equation:DPPH scavenging activity (%) = (A0 − A1)/A0 × 100
where A0 is the absorbance of the control without extract or positive control, and A1 is the absorbance of the sample.

### 4.5. Radical 2,2′-Azino-bis-3-ethylbenzthiazoline-6-sulphonic acid Cation (ABTS^•+^) Scavenging Activity

The ability of the acai extracts to scavenge ABTS^•+^ was determined according to the procedure of Acharya (2017) [88], with some modifications. Briefly, ABTS (7 mM) and potassium persulfate (2.45 mM) solutions were prepared in distilled water. A working solution was then prepared by combining 3 mL of each stock solution and letting them react for 12–16 h in the dark at room temperature (25 °C). The interaction of ABTS with potassium persulfate led to the formation of the ABTS radical cation (ABTS^•+^) [88]. The solution was then diluted by mixing 1 mL ABTS radical solution with 25 mL of PBS to obtain an absorbance of 0.70 at 750 nm, as monitored using a Varioskan™ LUX multimode microplate reader (ThermoFisher, Stafford, UK). A total of 200 µL reaction mixture per well was assessed, comprised of 190 µL of radical solution followed by 10 µL of standard or plant extracts at a concentration range of 1–1000 µg/mL. The plate was shaken for 10 s at medium speed and incubated for 5 min in the dark. Then, the absorbance was measured at 750 nm. The activity of the acai extracts was compared with other antioxidant standards, i.e., L-ascorbic acid, α-tocopherol, and gallic acid. The ability of the extracts to scavenge ABTS^•+^ was calculated using the following equation:ABTS^•+^ scavenging activity (%) = (A0 − A1)/A0 × 100
where A0 is the absorbance of the control, and A1 is the absorbance of the sample.

### 4.6. Hydrogen Peroxide (H_2_O_2_) Scavenging Activity

Humans are exposed to H_2_O_2_ either directly through mitochondrial metabolism or indirectly from the environment. H_2_O_2_ is widely considered a cytotoxic agent, and the rapid breakdown of H_2_O_2_ can produce a ^•^OH that can initiate lipid peroxidation and cause protein and DNA damage [89]. The H_2_O_2_ scavenging ability of the acai extracts was measured using the method of Alam et al. (2013) [90], with modifications. In 200 µL total solution per well of a 96-well plate, 180 µL of 40 mM H_2_O_2_ solution prepared in PBS was added, followed by 20 µL of standard or acai extracts at a concentration range of 1–4000 µg/mL. The mixture was incubated for 10 min, and the absorbance was read at 230 nm using a Varioskan™ LUX multimode microplate reader (ThermoFisher, Stafford, UK). The following equation was used to calculate the percentage of H_2_O_2_ scavenging:H_2_O_2_ scavenging activity (%) = (A0 − A1)/A0 × 100
where A0 is the absorbance of the control, and A1 is the absorbance of the sample.

### 4.7. Hydroxyl Radical (^•^OH) Scavenging Activity

The acai extracts’ ^•^OH scavenging activity was evaluated using the method described by Bajpai et al. (2015) [91], with modifications. The principle of this experiment is based on a Fenton’s reaction, which involves the Fe^3+^–ascorbate–ethylenediaminetetraacetic acid–H_2_O_2_ system to produce hydroxyl radicals. In a total volume of 200 µL, the reaction mixture contained 50 µL of 2-deoxy2-ribose sugar (12 mM), 20 µL of ferric chloride (FeCl_3_) (1 mM), 20 µL of ethylenediaminetetraacetic acid (EDTA) (1 mM), 20 µL of L-ascorbic acid (1 mM), 50 µL of H_2_O_2_ (8 mM), 30 µL of PBS, and 10 µL of standard or acai extracts at a concentration range of 1–4000 µg/mL. A volume of 40 µL of 2.8% trichloroacetic acid (TCA) and 2-thiobarbituric acid (TBA) (0.5% in 0.025 M sodium hydroxide solution) was added to the reaction mixture after 45 min at 37 °C, and the mixture was incubated at 85 °C for 15 min to generate a pink chromogen that resulted from the reaction of TBA with degraded sugar, i.e., a ‘malondialdehyde-like’ compound [92]. After cooling, 200 µL of the sample was transferred to a 96-well microtiter plate, and the absorbance was measured at 532 nm using a Varioskan™ LUX multimode microplate reader (ThermoFisher, Stafford, UK). Gallic acid was used as a reference standard. The percentage of inhibition activity was determined using the same formula as for the DPPH radical scavenging activity (Section 4.4).

### 4.8. Nitric Oxide Radical (^•^NO) Scavenging Activity

The procedures described by Unuofin et al. (2018) and Jimoh et al. (2019) [93,94] were adapted for the determination of the capability of the acai extracts to scavenge ^•^NO radicals. ^•^NO radicals can be produced by sodium nitroprusside (SNP) decomposing in an aqueous solution at pH (7.2) [90]. The Griess reagent can be used to determine ^•^NO quantities under aerobic conditions as NO reacts with oxygen to produce nitrates [90]. Briefly, 2 mL of 10 mM SNP in PBS was combined with 0.5 mL of acai extracts or butylated hydroxyanisole (BHA) at concentrations of 0.1–500 µg/mL. After 150 min of incubation at 25 °C, 0.5 mL of the solution was combined with 0.5 mL of Griess reagent, which was prepared by mixing 1 mL of 0.33% sulphanilamide reagent (in 20% glacial acetic acid) and 1 mL of 0.1% naphthalene diamine dichloride at room temperature for 5 min. Following a 30 min incubation period at room temperature, 150 µL of the mixture was transferred to a 96-well plate, and the absorbance was measured at 540 nm using a Varioskan™ LUX multimode microplate reader (ThermoFisher, Stafford, UK). A negative control was prepared using a water-based solution instead of the extract or standard BHA. Using the same formula as the DPPH radical scavenging activity (Section 4.4), the percentage of the nitric oxide scavenging activity was calculated.

### 4.9. Lipid Peroxidation (LPO) Inhibitory Activity

Reactive oxygen species (ROS) such as O^2•−^ anion, ^•^OH, and the H_2_O_2_ radical, trigger LPO, which damages cell membranes and produces numerous secondary products that are neurotoxic, resulting in neuronal death via necrosis or apoptosis [95]. Moreover, it has been shown that LPO contributes to the development of many NDDs, including AD [95]. The ability of acai extracts to inhibit LPO was assessed using a method modified from that described by Akomolafe et al. (2013) [96]. Briefly, 100 µL of 5 mg/mL bovine brain extract type I, Folch fraction I (Sigma, B1502) was mixed with 30 µL of PBS, 40 µL of distilled water, and 100 µL of acai extracts or standard at a concentration range of 0.1–1000 µg/mL, with 100 µL of 5 mM SNP as the prooxidant. After a 2 h incubation at 37 °C, 300 µL of 8.1% sodium dodecyl sulphate (SDS), 500 µL of acetic acid, and 500 µL of 0.8% thiobarbituric acid (TBA) were added. This mixture was incubated at 85 °C for 45 min to induce the formation of the malondialdehyde (MDA) coloured product. A volume of 200 µL of the samples was transferred to a 96-well microtiter plate after cooling, and the absorbance was measured at 532 nm using a Varioskan™ LUX multimode microplate reader (ThermoFisher, Stafford, UK). The percentage inhibition of the formation of MDA was calculated according to the equation for the DPPH radical scavenging activity (Section 4.4).

### 4.10. Ferric-Reducing Antioxidant Power (FRAP) Assay

The ability to reduce ferric ions (Fe^3+^) to ferrous ions (Fe^2+^) was used to estimate the reducing capacity of acai extracts. The acai extract concentrations were assessed over a concentration range of 0.001–8000 µg/mL. Each assay data point contained 4 µL of each acai extract, 400 µL of phosphate buffer (Gibco™ PBS, pH 7.4, ThermoFisher, Stafford, UK), and 250 µL of 1% potassium ferricyanide. After the incubation of the mixture at 50 °C for 20 min, 250 µL of 10% trichloroacetic acid was added. The samples were centrifuged at 3000 rpm for 10 min. Then, 100 µL of the supernatant was transferred to a 96-well microtiter plate and mixed with 100 µL of double-distilled water and 20 µL of freshly prepared (0.1%) ferric chloride (FeCl_3_) solution. Then, the formation of Perl’s Prussian blue was read at 700 nm, according to Nwidu et al. (2018) [82] using a Varioskan™ LUX multimode microplate reader (ThermoFisher, Stafford, UK). The positive control was L-ascorbic acid.

### 4.11. Total Phenolic Content Determination

Based on the Folin–Ciocalteu reagent (FCR) method, the total phenolic content was determined spectrophotometrically at 760 nm according to Nwidu et al. (2018) [82]. In this assay, electrons were transferred from phenolic compounds to phosphomolybdic/phosphotungstic acid complexes (Folin–Ciocalteu reagent) under alkaline conditions, resulting in a detectable colour change [97]. A concentration range of 15.63–3000 µg/mL was used to evaluate the acai extracts. Each assay data point within a 96-well plate contained 20 µL of acai extract, 90 µL of water, and 30 µL of FCR; then, the mixture was shaken vigorously in a plate reader for 8 min. Then, 60 µL 7.5% sodium carbonate solution was added, and the plate was incubated at 40 °C on a shaking incubator for 30 min. The plate was read in a spectrophotometer at 760 nm using a Varioskan™ LUX multimode microplate reader (ThermoFisher, Stafford, UK). The positive control was gallic acid, using a concentration range of 15.63–1000 µg/mL to generate a standard curve for the quantification of the total phenolic content of the acai extracts. The total phenolic content was determined as mg gallic acid equivalents/gram of plant extract (mg GAE/g).

### 4.12. Total Flavonoid Content Determination

This colourimetric method is based on the principle that aluminium chloride (AlCl_3_) forms acid-stable complexes with flavone and flavonol keto groups and their hydroxyl groups [98]. Furthermore, AlCl_3_ produces acidic compounds with orthodihydroxyl groups in flavonoid A- or B-rings [98]. The total flavonoid contents of the plant extracts were assessed via the colourimetric procedure described in Nwidu et al. (2018) [82]. The positive control was quercetin. Within a 96-well plate, 20 µL of acai extracts or quercetin as standard, over a concentration range of 15.63–3000 µg/mL, was mixed with 100 µL of 10% aluminium chloride solution and 100 µL 1 M potassium acetate. After a 30 min incubation at room temperature, the plate was read using a Varioskan™ LUX multimode microplate reader (ThermoFisher, Stafford, UK) at 415 nm. Total flavonoid content was expressed as milligram quercetin equivalents/gram of extract (mg QUER E/g).

### 4.13. Statistical Analysis

Results were expressed as means ± standard error of the mean (SEM) in each treatment and control group. Non-linear regression analysis was used to calculate the concentration of acai extracts producing 50% inhibition (IC_50_). The statistical analysis comparing different groups was performed using one-way ANOVA tests with Tukey’s multiple comparisons post-test via PRISM v7 (GraphPad Software Inc., San Diego, CA, USA. www.graphpad.com, accessed on 26 June 2022). A *p*-value of below 0.05 was defined as the level of statistical significance for all analyses.

## 5. Conclusions

To summarise, the acai berry contains a range of phytochemicals that likely contribute to its anti-cholinesterase and antioxidant activities. This study suggests that the acai aqueous extract could be further fractionated, and its compounds identified for their potential use as a medication alternative for AD therapy, due to the potent ChEI activity and powerful antioxidant capabilities. However, the limitation of this study is that the work was performed in vitro; future in vivo analyses are required, particularly those that mimic NDDs such as AD.

## Figures and Tables

**Figure 1 molecules-27-04891-f001:**
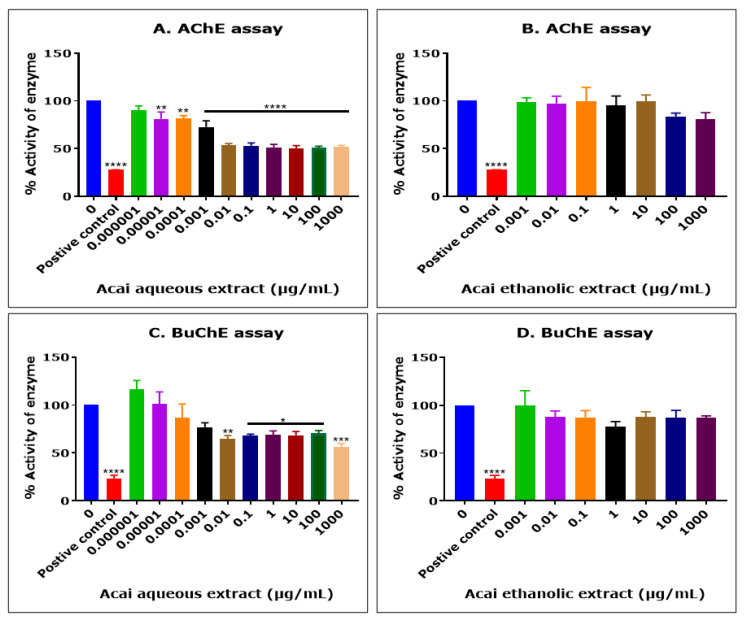
Cholinesterase inhibition by acai aqueous and ethanolic extracts. AChE inhibitory activity of acai aqueous extract (**A**) and acai ethanolic extract (**B**). BuChE inhibitory activity of acai aqueous extract (**C**) and acai ethanolic extract (**D**). Histograms represent means ± SEM for at least three replicate assays at each extract concentration (*n* = 6). For the positive control inhibitors, 5 mM azamethiphos and 5 mM ethopropazine hydrochloride were used for AChE and BuChE, respectively. For marked significance * *p* < 0.05, ** *p* < 0.01, *** *p* < 0.001, and **** *p* < 0.0001.

**Figure 2 molecules-27-04891-f002:**
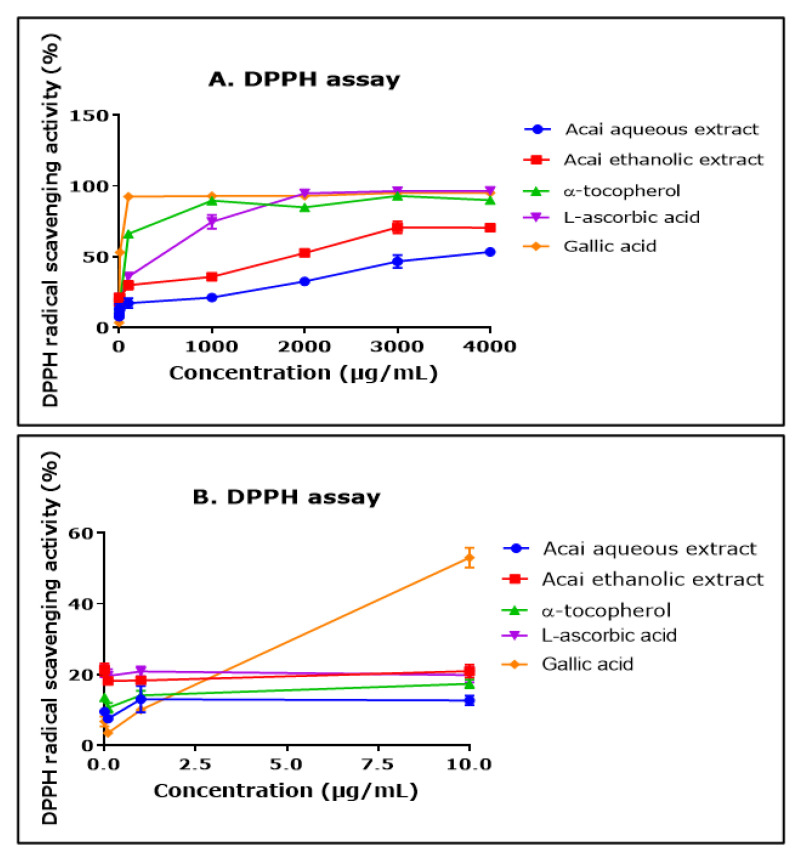
DPPH radical scavenging activity of acai aqueous and ethanolic extracts. Acai antioxidant activity was assessed via the percentage inhibition (radical scavenging) of DPPH over a concentration range of 0.01–4000 µg/mL (**A**) and 0.01–10 µg/mL (**B**). Assays were performed in triplicate at each extract concentration (*n* = 6).

**Figure 3 molecules-27-04891-f003:**
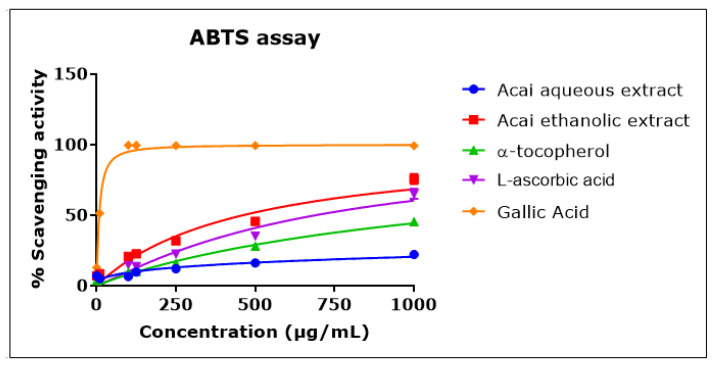
ABTS^•+^ scavenging activity of acai aqueous and ethanolic extracts. Acai antioxidant activity was assessed as a percentage inhibition (radical scavenging) of ABTS over a concentration range of 1–1000 µg/mL. Assays were performed in triplicate at each extract concentration (*n* = 6).

**Figure 4 molecules-27-04891-f004:**
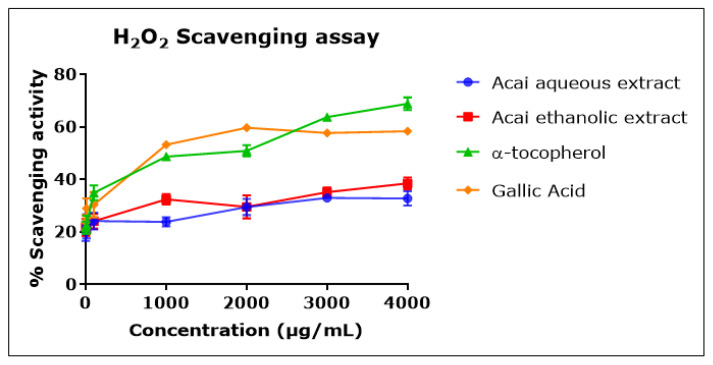
H_2_O_2_ scavenging activity of acai aqueous and ethanolic extracts. Acai antioxidant activity was assessed as a percentage of the scavenging activity of H_2_O_2_ over a concentration range of 1–4000 µg/mL. Assays were performed in triplicate at each extract concentration (*n* = 6).

**Figure 5 molecules-27-04891-f005:**
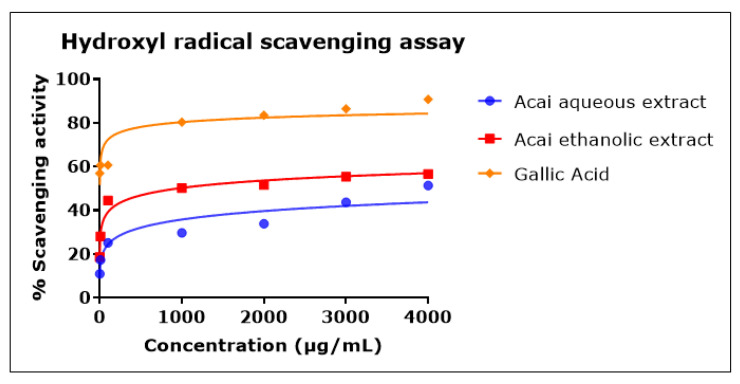
Hydroxyl radical scavenging activity of acai aqueous and ethanolic extracts. Acai antioxidant activity was assessed via the percentage of the scavenging of ^•^OH. Assays were performed in triplicate at each extract concentration (*n* = 6).

**Figure 6 molecules-27-04891-f006:**
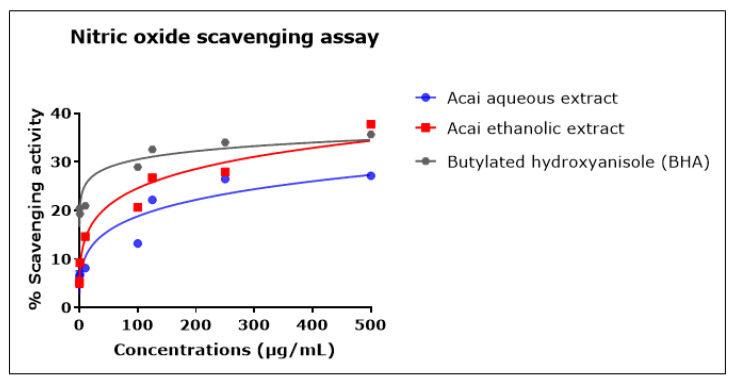
Nitric oxide (^•^NO) scavenging activity of acai aqueous and ethanolic extracts. Acai antioxidant activity was assessed via the percentage of the scavenging of ^•^NO. Assays were performed in triplicate at each extract concentration (*n* = 6).

**Figure 7 molecules-27-04891-f007:**
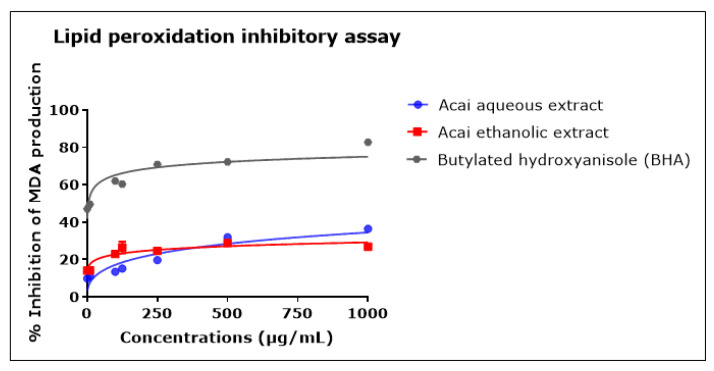
Lipid peroxidation inhibitory activity of acai aqueous and ethanolic extracts. Acai antioxidant activity was assessed via the percentage of the inhibition of malondialdehyde (MDA) production. Assays were performed in triplicate at each extract concentration (*n* = 6).

**Figure 8 molecules-27-04891-f008:**
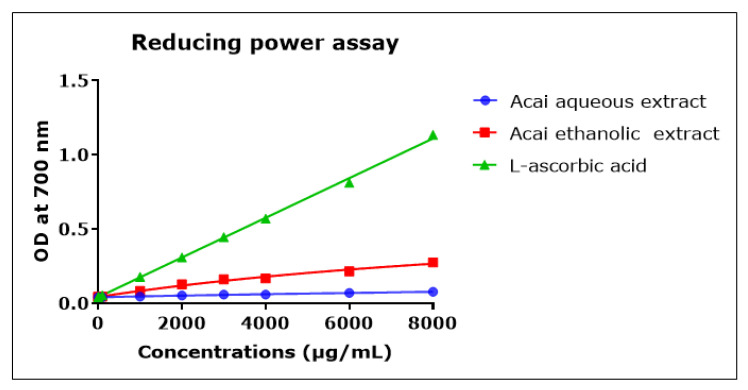
Reductive capacity of different concentrations of plant extracts from acai berry. Plant reducing power was measured by the ability to reduce ferric (Fe^3+^) to ferrous (Fe^2+^) iron (OD density change at 700 nm). The positive control was L-ascorbic acid (vitamin C). Assays were performed in triplicate at each extract concentration (*n* = 6).

**Table 1 molecules-27-04891-t001:** The approximate IC_50_ values (mg/mL) of acai aqueous extract, acai ethanolic extract, α-tocopherol (vitamin E), L-ascorbic acid (vitamin A), gallic acid, and butylated hydroxyanisole (BHA) for the AChE, BuChE, DPPH, ABTS, H_2_O_2_, ^•^OH, ^•^NO, and LPO assays.

Sample	AChE	BuChE	DPPH	ABTS	H_2_O_2_	^•^OH	^•^NO	LPO
Acai aqueous extract	0.014	6.378	11.550	30.541	7.803	11.604	12.932	4.862
Acai ethanolic extract	NS	NS	0.791	0.462	1.479	0.946	4.544	438.8
α-tocopherol	-	-	0.050	1.270	0.676	-	-	-
L-ascorbic acid	-	-	0.115	0.690	-	-	-	-
Gallic acid	-	-	0.008	0.008	0.737	0.001	-	-
Butylated hydroxyanisole	-	-	-		-	-	135.437	0.004

(-), Not evaluated; NS, not significant.

**Table 2 molecules-27-04891-t002:** Total phenolic (TPC) and flavonoid (TFC) contents of acai berry aqueous and ethanolic extracts.

Acai Berry Extracts	Total Phenolic Content (mg GAE/g)	Total Flavonoid Content (mg QUER E/g)
Acai aqueous extracts	19.42 ± 0.40	1.26 ± 0.11
Acai ethanolic extracts	101.39 ± 4.61	11.78 ± 1.42

Values represent means ± standard error of the mean (SEM) of triplicate assays (*n* = 6). GAE: gallic acid equivalent; mg QUER E/g: milligram quercetin equivalents/gram of extract.

## Data Availability

Data supporting the results are available on request from the first author (M.N.A.).

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
