# Peer review of "A Preliminary Assessment of the Nutraceutical Potential of Acai Berry (Euterpe sp.) as a Potential Natural Treatment for Alzheimer’s Disease"

_molecules, 2022, doi:10.3390/molecules27154891_

Round 1

Reviewer 1 Report

The present study evaluated the nutraceutical of acai berry (Euterpe sp.) as a potential natural treatment for Alzheimer’s disease. The authors quantify antioxidant potential via hydrogen peroxide, hydroxyl scavenging, nitric oxide scavenger, lipid peroxidation inhibition, and the ability to reduce ferric ions and anti-cholinesterase capability by inhibiting acetyl- and butyryl-cholinesterase enzymes.

In the materials and methods section,  the authors show starting weight of acai berry (300 mg/mL) for ethanolic extraction but not for aqueous extraction. The author must show starting weight for aqueous extraction.

Different concentrations of both extracts are used for quantifying antioxidant potential and anti-cholinesterase capability. The author must explain why they used different concentrations for quantifying antioxidant potential and anti-cholinesterase capability?

The authors quantify phenolic compounds and flavonoid contents in ethanolic and aqueous extract. Antioxidant properties have been widely discussed in phenolic compounds and flavonoid contents. However, it is unclear which molecules mainly found in acai berry can exert the effects showed in the extracts.

1.      The chemical composition and compound distribution of acai berry must be showed.

2.      The antioxidant and anti-cholinesterase potential of those molecules should be studied in silico by molecular docking.

Author Response

The present study evaluated the nutraceutical of acai berry (Euterpe sp.) as a potential natural treatment for Alzheimer’s disease. The authors quantify antioxidant potential via hydrogen peroxide, hydroxyl scavenging, nitric oxide scavenger, lipid peroxidation inhibition, and the ability to reduce ferric ions and anti-cholinesterase capability by inhibiting acetyl- and butyryl-cholinesterase enzymes.

In the materials and methods section,  the authors show starting weight of acai berry (300 mg/mL) for ethanolic extraction but not for aqueous extraction. The author must show starting weight for aqueous extraction.

We thank Reviewer #1 for spotting this oversight, the starting weight for aqueous extraction should have been included.  We have amended the text of the revised manuscript to include the starting weight of the aqueous extract (10 mg/mL).

Different concentrations of both extracts are used for quantifying antioxidant potential and anti-cholinesterase capability. The author must explain why they used different concentrations for quantifying antioxidant potential and anti-cholinesterase capability?

The acai aqueous and ethanolic extracts were assayed over the concentration range of 1 x10-6 µg/mL to 1000 µg/mL for anti-cholinesterase activity.  The concentration range employed simply reflects one that was below the threshold of activity (for example, 1 x10-6 µg/mL of the aqueous extract was without effect) and covered the maximal effects (a plateau of AChE inhibition was observed between 0.01-1000 µg/mL).  By contrast, antioxidant potential did not always reach saturation at 1000 µg/mL.  The saturation of lipid peroxidation and the ABTS assays were observed at 1000 µg/mL, hence this was the maximal data point.  However, other assays reached saturation at slightly lower concentrations (nitric oxide scavenging at 500 µg/mL) or a higher range (4000 µg/mL for DPPH radical scavenging, hydrogen peroxide scavenging, and hydroxyl radical scavenging). Reducing power (ferric to ferrous iron reduction) was still linear for the positive control antioxidant (L-ascorbic acid) even up to 8000 µg/mL, hence the extended range for the test extracts.  The concentration range used for the aqueous and ethanolic extracts for anti-cholinesterase activity was in part covered in the Figure Legend of Figure 1, but we appreciate that this should also have been included in the Methods section (4.3) and have therefore added these details to the revised manuscript. 

The authors quantify phenolic compounds and flavonoid contents in ethanolic and aqueous extract. Antioxidant properties have been widely discussed in phenolic compounds and flavonoid contents. However, it is unclear which molecules mainly found in acai berry can exert the effects showed in the extracts.

  1. The chemical composition and compound distribution of acai berry must be showed.

A number of publications have considered the chemical composition of the acai berry including the presence of numerous polyphenols and flavonoids such as anthocyanins and we briefly cover this in the discussion section and cite appropriate references [50,74-76,78,79]. To extend this further, we have generated a new Table (Supplementary Table S1) that covers the chemical composition and reported anti-cholinesterase and antioxidant activities.  As mentioned in the discussion section, the basis for a future study would be to resolve the extracts by chromatographic means and isolate chemical components and then assay these to specifically identify the major agent(s) responsible for the anti-cholinesterase and antioxidant activities, but this itself will be a separate, stand-alone publication.

  1. The antioxidant and anti-cholinesterase potential of those molecules should be studied in silico by molecular docking.

We appreciate the suggestion by Reviewer #1 to follow up this work with in silico modelling to better understand which molecules specifically bind to the active or peripheral binding sites of AChE and BuChE.  We have undertaken a similar in silico approach after identification (by LC-MS/MS) of some of the compounds present in the leaves from several commonly cultivated plants (Amat-Ur-Rasool et a. 2020. Potential Nutraceutical Properties of Leaves from Several Commonly Cultivated Plants. Biomolecules. 2020 Nov 15;10(11):1556. doi: 10.3390/biom10111556).  However, there are a plethora of compounds that may be responsible for the anti-cholinesterase and antioxidant activities (refer to the new Supplementary Table S1), and therefore to model these individually would not be feasible.  As discussed above, we would hope to optimize a chromatographic strategy to enable isolation, and then identification and characterization of the individual anti-cholinesterase and antioxidant compounds present within the acai berry extracts, but this represents a separate, stand-alone paper.

Reviewer 2 Report

I suggest to authors to change the type of paper from "Article" to "Communication" and add in the title "Preliminary Results"

In Introduction before line 72, the authors should add some lines on the importance of use of bioactive compounds from nutraceutical to medical fields and the innovative directions of nanonutraceuticals and related references should be mentioned such as: 

Yeung et al.  2020. Big impact of nanoparticles: analysis of the most cited nanopharmaceuticals and nanonutraceuticals research. Current Research in Biotechnology, 2, 53 – 63, 2020. DOI: 10.1016/j.crbiot.2020.04.002.

The authors should better describe the profile of bioactive components of acai berry.

A graphical scheme of study approach should be inserted in Material and Methods. Information on type and number of samples and samplings should be given.

In Material and Methods additional information on evaluation of antioxidant properties approach should be given. 

Resolution of Figure 1 should be improved.

Results on Hydroxyl radical scavenging activity and •NO scavenging should be better described in the text.

Data in figure 7 should be better described in the text.

Data in Table 1 should be better described in the text.

Lines 290-402 should be better explained.

Lines 426-437 should be better explained.

Limits, advantages, practical applications, and future directions should be added in Conclusion.

Author Response

I suggest to authors to change the type of paper from "Article" to "Communication" and add in the title "Preliminary Results"

We thank Reviewer #2 for this suggestion and appreciate that the results are preliminary and, as covered in the discussion, further identification and characterization of the active agent(s) responsible for the antioxidant and anti-cholinesterase activities will be needed in a follow-up manuscript.  We have therefore amended the title of the manuscript to include “preliminary assessment of the…”.

In Introduction before line 72, the authors should add some lines on the importance of use of bioactive compounds from nutraceutical to medical fields and the innovative directions of nanonutraceuticals and related references should be mentioned such as: 

Yeung et al.  2020. Big impact of nanoparticles: analysis of the most cited nanopharmaceuticals and nanonutraceuticals research. Current Research in Biotechnology, 2, 53 – 63, 2020. DOI: 10.1016/j.crbiot.2020.04.002.

We thank Reviewer #2 for this suggestion and have included a brief introduction to nutraceuticals and their growing importance in medicine and as potential treatments to combat disease pathogenesis and/or progression in the introduction of the revised manuscript. New references have been cited and included in the revised manuscript.  However, the specific paper suggested by Reviewer #2 is useful but focused on nano-delivery of pharmaceuticals and nutraceuticals and is therefore not relevant to our study. The focus of our manuscript is that of characterization of bioactivities within the acai berry in keeping with the remit of this Special Issue and does not consider specific delivery mechanisms for nanopharmaceuticals or nanonutraceuticals.

The authors should better describe the profile of bioactive components of acai berry.

We thank the suggestion from Reviewer #2 and have generated a new Supplementary Table S1 that details known bioactive components of the acai berry and their potential antioxidant and anti-cholinesterase properties.

A graphical scheme of study approach should be inserted in Material and Methods. Information on type and number of samples and samplings should be given.

We thank Reviewer #2 for this suggestion.  We have now generated a graphical abstract that covers the methods employed. The number of experiments for each of the assays has also been added to the Figure Legends of the revised manuscript.

In Material and Methods additional information on evaluation of antioxidant properties approach should be given. 

We thank Reviewer #2 for this suggestion.  We have added text to the revised manuscript that provides additional methodological detail on the basis of each of the antioxidant assays and their importance.

Resolution of Figure 1 should be improved.

The resolution of this Figure reflects the sizing of this Figure.  By increasing the size, the clarity of the image has been restored. In the finalized manuscript, high-resolution images will be submitted separately to the publishers.

Results on Hydroxyl radical scavenging activity and •NO scavenging should be better described in the text.

The results for both of these analyses are described in the text and to a similar extent as the other assays.  As detailed above, we have added more detail to the description of the methods and included n-numbers in the Figure legends for both.

Data in figure 7 should be better described in the text.

The data in Figure 7 are described to a similar level as those for the other assays. We appreciate that more description of the relevance of a lipid peroxidation assay could be included, and this has been added to the methods section of the revised manuscript.

Data in Table 1 should be better described in the text.

The data in Table 1 are IC50 values only from each of the assays for reference.

Lines 290-402 should be better explained.

As detailed above, we have added more text to fully explain the methodology and rationale for the assays.

Lines 426-437 should be better explained.

As detailed above, we have added more text to fully explain the methodology and rationale for the assays.

Limits, advantages, practical applications, and future directions should be added in Conclusion.

This journal does not have a conclusion section.  We have added detail about the limitations of the work and the future studies that are required in the final paragraph of the discussion section.

Round 2

Reviewer 1 Report

The authors improved manuscript.